# Power Line Scene Recognition Based on Convolutional Capsule Network with Image Enhancement

**Kuansheng Zou \*, Shuaiqiang Zhao and Zhenbang Jiang**

School of Electrical Engineering and Automation, Jiangsu Normal University, Xuzhou 221116, China
\* Correspondence: zoukuansheng@jsnu.edu.cn

**Abstract:** With the popularization of unmanned aerial vehicle (UAV) applications and the continuous development of the power grid network, identifying power line scenarios in advance is very important for the safety of low-altitude flight. The power line scene recognition (PLSR) under complex background environments is particularly important. The complex background environment of power lines is usually mixed by forests, rivers, mountains, buildings, and so on. In these environments, the detection of slender power lines is particularly difficult. In this paper, a PLSR method of complex backgrounds based on the convolutional capsule network with image enhancement is proposed. The enhancement edge features of power line scenes based on the guided filter are fused with the convolutional capsule network framework. First, the guided filter is used to enhance the power line features in order to improve the recognition of the power line in the complex background. Second, the convolutional capsule network is used to extract the depth hierarchical features of the scene image of power lines. Finally, the output layer of the convolutional capsule network identifies the power line and non-power line scenes, and through the decoding layer, the power lines are reconstructed in the power line scene. Experimental results show that the accuracy of the proposed method obtains 97.43% on the public dataset. Robustness and generalization test results show that it has a good application prospect. Furthermore, the power lines can be accurately extracted from the complex backgrounds based on the reconstructed module.

**Keywords:** capsule network; image enhancement; power line scene recognition; complex background

## 1. Introduction

With the continuous development of the modern power grid system, the demand for electricity is also greatly increased, and transmission lines spread to all parts of the world in a complex network. It is also of great significance for low-altitude flight to detect the power lines and implement obstacle avoidance. The Australian transport safety report shows that between 1994 and 2004, there were 119 helicopter crashes into power lines, of which 45 caused fatal injuries and 22 caused serious injury [1]. Hitting power lines will cause serious damage to the helicopter. The U.S. military data report shows that 54 power line collisions occurred between 1997 and 2006, resulting in 13 deaths and economic losses of up to USD 224 million [2]. Flight safety accidents threaten people's lives and cause huge economic losses.

Flight obstacle avoidance mainly depends on the pilot's reaction and experience. They can avoid large obstacles, but small obstacles, especially power lines, they often fail to dodge, which in turn leads to disasters. The power line scene recognition (PLSR) is mainly used for the flight obstacle avoidance of power lines, which can identify the presence or absence of power lines in advance, and use this as a judgment basis for reminding the driver. Thus, it is a meaningful research work and has a huge market prospect.

Although there were many publications in scene recognition of remote sensing images [3–8], little research focused on the PLSR. The leading cause for this is that the public dataset of the power lines is very scarce; only three types of power line data sets could

be easily downloaded on the internet [9–12]. Among them, only one type can be used for the classification and recognition of power line scenarios [9,10]. Due to the inherent characteristics of power lines and the low resolution of datasets, it is difficult to obtain good recognition results. There were still state-of-the-art PLSR methods presented in recent years. Yetgin et al. [13] presented a PLSR framework based on the discrete cosine transform (DCT) of scenes obtained from aircraft-based cameras. This work attacked the problem of extracting signatures from the DCT coefficients by systematically changing the DCT matrix sizes and applying known classifiers to the DCT sub-matrices. The details were given in [14]. First, the image filtering was used to reduce the interference of noise and normalize the amplitude. Second, different types of image features of power lines were extracted through the DCT, local binary pattern (LBP), and histogram of oriented gradient (HOG), respectively. The absolute value of the logarithm of the discrete cosine coefficient in the DCT domain was taken to emphasize the dynamic range. Finally, the naive bayes (NB), random forest (RF), and support vector machine (SVM) classifier were used for the PLSR task. Although these kinds of methods were simple, it needed to manually set the feature extraction and feature matching methods. The PLSR method, based on deep learning, does not require manual feature extraction of power lines, and the established convolutional neural network (CNN) model can automatically extract effective features. Thus, some researchers tried to apply the CNNs to PLSR [15]. The VGG19 model and the ResNet50 model were fine-tuned to adapt to the power line dataset in literature [15], and an end-to-end PLSR method is proposed. The VGG19 model and the ResNet50 model were divided into five stages, and then the feature maps of these five stages were outputted. The feature maps were inputted to the NB, RF, and SVM classifiers, respectively, for the PLSR task. A fast PLSR network for the pixel-wise straight and curved power line detection method is proposed in [16]. The edge attention fusion module was combined together with a filter block, which extracts semantic and spatial information to improve the PLSR result along the boundary.

The power line extraction (PLE) is the pixel-wise PLSR method, which was paid more attention than the PLSR task. A PLE method based on the weakly supervised learning, which solved the problem of labeling large-scale datasets, was proposed in [17]. A PLE method based on pyramid patch classification, which used a CNN-based classifier to help eliminate power line pseudo-targets, was proposed in [18]. The generative adversarial network was combined with the conic and hue perturbation to enhance the datasets to reduce the model parameters and computational complexity through model pruning in [19]. Artificially synthesized power line images were used as the training data, and a fast single-shot line segment detector (LSD) was proposed in [20]. A real-time segmentation model for power lines was proposed in [21]. They used a spatial branch to capture rich spatial information and utilized classification with subnet-level skip connections. It recovered long-distance features and improved the performance of the power line extraction. Liu et al. improved the Unet model and its variants to the power line scene recognition and extraction task [22].

Since the capsule network (CapsNet) [23] is widely used in various classification tasks with its rich feature expression ability and effectiveness on small data sets and achieved good classification results [24–29]. The CapsNet is also tentatively studied in the scene recognition of remote sensing images [3–8]. Thus, in this paper, the CapsNet is selected as the backbone network, and the edge of line features of the power lines are enhanced. Finally, a novel PLSR method is proposed. The main innovation can be summarized as follows:

(1) A PLSR method based on the convolutional CapsNet fused with image enhancement is proposed. The edge structures of the power lines are enhanced by using the guided filter. The lone points and lines that are reinforced at the same time are weakened by the convolutional CapsNet. Various experiments show that it is suitable for the PLSR task with complex backgrounds.

(2) The power line scene recognition and feature extraction tasks can be performed simultaneously based on the convolutional CapsNet structure. The PLSR task is performed based on the output of the digital capsule layer, and the PLE task is performed based on

the output of a reconstructed module. The sections of this paper are arranged as follows: The CapsNet is introduced in Section 2. The proposed convolutional CapsNet with image enhancement is explained in Section 3. The scene recognition results and analysis of the proposed method are given in Section 4. The reconstruction results and analysis of the proposed method are shown in Section 5. The conclusion is obtained in Section 6.

## 2. Capsule Network

The CapsNet is used to maintain the location information and the inherent attributes of objects in the image, which can model the spatial relationship of the image [23]. In the CapsNet structure, the scalar output of the feature detector in the CNN is replaced with a vector output, and the maximum pooling is replaced with a protocol routing simultaneously. Meanwhile, all the capsules, except the last capsule layer, maintain the convolutional structure. By doing this, the advantages of the CNN in copying the learned knowledge across space is retained. The higher-level capsules can cover a larger image area the same as the CNN. Unlike with the maximum pooling, the CapsNet can partially retain the precise location information of entities in the region through the protocol routing [30]. The CapsNet is composed of the input layer, output layer, convolutional layer, primary caps layer, and digit caps layer. The convolutional layer is used to extract the low-level features of the detect target. The primary caps layer is used to express the spatial relationship between the features, and transfers the extracted features to the digit caps layer. The dynamic routing algorithm is used to predict the classification results in the digit caps layer [31]. The coupling coefficient $c$, according to the similarity between the low-level capsule layer and the high-level capsule layer, is adjusted. The weight $W$ between networks is updated. If the similarity between the $i$-th capsule in the lower layer and the $j$-th capsule in the upper layer is greater, the coupling coefficient $c_{ij}$ is greater, and the formula is shown in Equation (1). Where the initial value of a priori coupled probability $b_{ij}$ of the capsule $i$ and the capsule $j$ is set to 0, and updated as Equation (2).

$$c_{ij} = \frac{\exp(b_{ij})}{\sum_k \exp(b_{ij})} \tag{1}$$

$$b_{ij} \leftarrow b_{ij} + \hat{u}_{j|i} v_j \tag{2}$$

where the calculation method is shown as Equations (3) and (4), respectively.

$$\hat{u}_{j|i} = W_{ij} u_i \tag{3}$$

$$v_j = \frac{\|s_j\|^2}{1 + \|s_j\|^2} \frac{s_j}{\|s_j\|} \tag{4}$$

where in Equation (4), $S_j$ is the input vector of the $j$-th capsule in the upper layer, and the formula is given as follows:

$$s_j = \sum_i c_{ij} \hat{u}_{j|i} \tag{5}$$

The object function of the CapsNet is defined as follows:

$$L_k = T_k \cdot max\left(0, \left(m^+ - \|v_k\|\right)\right)^2 + \lambda(1 - T_k) \cdot max\left(0, \left(\|v_k\| - m^-\right)\right)^2 \tag{6}$$

where $v_k$ is the output of a capsule in the softmax layer. $T_k$ represents the tag of the $k$-th target. If a training sample belongs to class $k$, $T_k = 1$. Otherwise, $T_k = 0$. $m^-$, and $m^+$ are, respectively, the upper bound for the probability of a training sample not belonging to class $k$ and the lower bound for the probability of a training patch being an instance of class $k$. They are set as $m^+ = 0.9$ and $m^- = 0.1$. $\lambda$ is a weight regularization factor, which is usually set as 0.5 [32].

The CapsNet was used to classify the MNIST images of $28 \times 28$ at first. The original network has a convolution layer, including 256 convolution cores with a scale of $9 \times 9$, and outputs a local feature map with a scale of $20 \times 20$ as the input of primary caps. The primary caps contain 32 different capsules, each with eight $9 \times 9 \times 256$ convolution kernels. Both layers use the ReLU activation function. Moreover, the digital capsule layer outputs 16-D vector reconstruction objects contain all the required instantiation parameters [26,33].

## 3. Convolutional Capsule Network with Image Enhancement

### 3.1. Motivation

The aerial image of power lines is mainly taken by the inspected unmanned aerial vehicles (UAVs), which has its inherent characteristics. In terms of color and lustre, the brightness of power lines is uniform and is higher than the backgrounds. In terms of shape, the power line usually exists in the form of a straight line, with a pixel width of about 1~5 [23], but some power lines, in the shape of a solitary vertical curve, still exist. In terms of spatial relationship, power lines usually run parallel to each other throughout the image, except for single ones. The background of power lines is complex and changeable. It is found that the background images of power lines are mostly forest, lake, river, field, mountain, sky, white cloud, pole, tower scene, and so on. It makes the power line scene recognition and extraction task challenging.

In general, power lines account for less than 15% of pixels in power line scenes. The complex backgrounds also have good edge features. Thus, pooling operation in the CNN may lose the spatial information of power lines, or misdetect part of the edge background as power lines. Due to the excellent performance of the CapsNet in the image classification mentioned above, the CapsNet is our first choice for the PLSR task. The CapsNet also has drawbacks: (1) It is unable to handle large size input well (2) It is unable to fully extract the input features. (3) The classification accuracy decreases with the complexity of the dataset. Two additional convolutional layers are used to better extract features and reduce input size simultaneously. The guided filter can enhance the edge lines well, meanwhile, the CapsNet can preserve the spatial relationship of power lines. Thus, the convolutional CapsNet with image hencement by guided filter is proposed.

### 3.2. Image Enhancement with Guided Filter

Experiments show that the guided filter [34] proposed by He et al. can better enhance the edge features of power lines and increase the recognition accuracy of power lines in complex backgrounds. The guided filter [34] is an edge-preserving algorithm based on the local linear model. It uses a guided image to guide the filtering process, defines any pixel in the image as a linear relationship with some of its adjacent pixels, and performs filtering processing, respectively. Finally, all local filtering results are accumulated to derive the global filtering results, and an output image with a structure similar to the input image is obtained.

The output image $f^o$ of the guided filter can be linearly represented by the guided image $I_i$ in a square window $i$, as shown below [35].

$$f^o = a_k I_i + b_k, \ \forall i \in w_k \tag{7}$$

where $w_k$ is a square window with a radius of $r$ centered on the pixel $k$, $a_k$ and $b_k$ are constants in $f^o$, and their coefficients are solved by minimizing the following energy function:

$$E(a_k, b_k) = \sum_{i \in w_k} \left( \left( f_i^o - f_i^{in} \right)^2 + \eta a_k^2 \right) \tag{8}$$

where $\eta$ is the regularization parameter to prevent it with too large a value, $f_i^{in}$ is the input image of the filter.

Because the guided filter uses a guided image for reference, choosing a different guided image will obtain different learning tasks. It is suitable for the deep learning

process. A power line scene image as input is shown in Figure 1a, and its enhanced image by the guided filter is shown in Figure 1b. Where the input image itself is selected as the guided image. It is obvious that the power lines are enhanced. Simultaneously, grass and the outline of a wheat field are also enhanced. If the CNN is used for the deep learning network, these enhanced backgrounds will represent the surrounding spaces because of several pooling operations. If the surrounding spaces are considered by the CapsNet, power lines can be easily distinguished with the enhanced backgrounds.

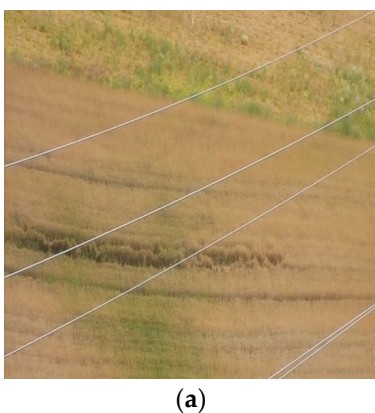 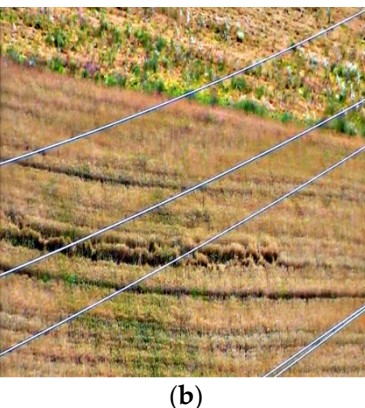

(**a**)   (**b**)

**Figure 1.** Power line scene image enhancement by using a guided filter. The input image itself is selected as the guided image. (**a**) Power line image. (**b**) Enhanced image.

If the ground truth images are selected as the guided image, the training process of the network will be sped up. A power line image as input is shown in Figure 2a, the ground truth label is shown in Figure 2b, and the output image of the guided filter by using the ground truth as the guided image is shown in Figure 2c. Obviously, the output image, by using the guided filter, is greatly enhanced. If this type of guided filter is combined with deep learning, not only will the training time be greatly reduced, the network performance will be also improved.

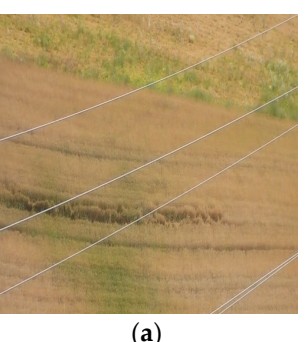 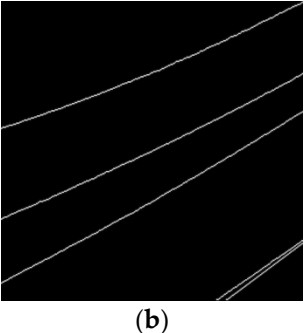 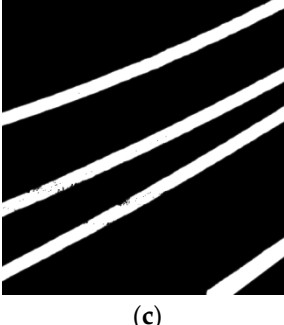

(**a**)   (**b**)   (**c**)

**Figure 2.** Power line scene image enhancement by using a guided filter. The ground truth is selected as the guided image. (**a**) Power line image. (**b**) Ground Truth. (**c**) Enhanced image.

In practice, there are no responded ground truth labels with the input images. Except for choosing the input image itself, more clarity for an image with special features can be selected as the guided image. For example, the line segment detection (LSD) [36] can better outline the power lines, it can be considered as the guided image. A power line image as input is shown in Figure 3a, the responded LSD map is shown in Figure 3b, and the output image of the guided filter by using the LSD as the guided image is shown in Figure 3c. Obviously, the output image, by using the guided filter, is greatly enhanced.

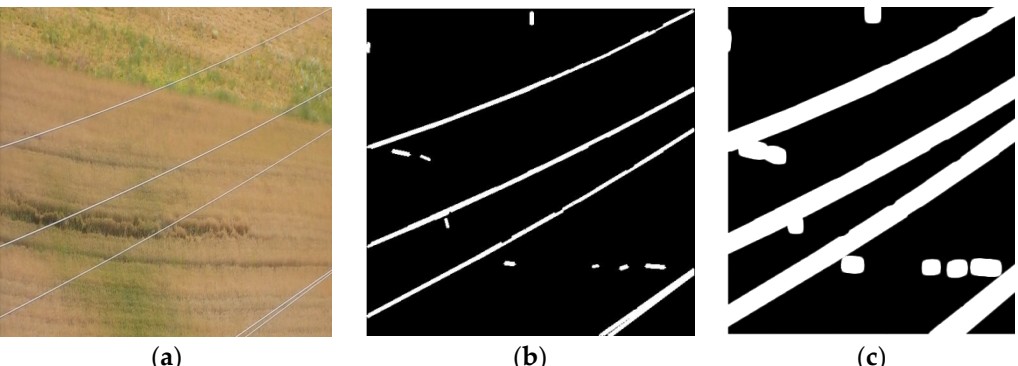

**Figure 3.** Power line scene image enhancement by using the guided filter. The LSD image is selected as the guided image. (**a**) Power line image. (**b**) LSD image. (**c**) Enhanced image.

In order to further improve scene recognition performance, the reconstructed image can be used as the guided image again for feedback. For feature extraction and classification, the ground truth image, feature enhanced image, or reconstructed image can be selected as the guided image to improve the accuracy. The input image itself is used as the guided image, and has a wider application. In order to make a more general comparison, the input image itself is selected as the guided image in the experiment. In brief, the guided filter with its variations [37–40], has a broad research prospect in the field of deep learning.

*3.3. Convolutional CapsNet Framework*

The proposed PLSR framework is shown in Figure 4. After image enhancement, the original image of 128 × 128 × 3 is grayed to an image of 128 × 128 × 1 and enters the first convolutional layer. The first convolutional layer contains 32 kernel functions with a scale of 5 × 5, and stride = 2. The output 64 × 64 × 32 feature image enters the second convolutional layer. The second convolutional layer contains 64 kernel functions with a scale of 5 × 5, stripe = 2. The output 32 × 32 × 64 feature map enters the third convolutional layer. The third convolutional layer contains 128 kernel functions with a scale of 9 × 9, and stripe = 2. The output 16 × 16 × 128 feature map enters the primary capsule layer. The primary capsule layer contains 32 different capsules, each capsule performs eight times of 9 × 9 kernel convolution, and stripe = 1. The last digital capsule layer outputs 16-D vector, which is used for binary classification tasks (power line scene or non-power line scene), and provides necessary information for image reconstruction. The ReLU activation function is applied to all layers. After the subsequent reconstruction module, the digital capsule can reconstruct the extracted power line binary image. The dimensions are 128 × 128 × 1.

The specific parameters of the convolutional CapsNet structure are shown in Table 1. In this paper, before the primary capsule layer, three convolutional layers with a stripe of 2 are selected in order to reduce the image dimension and extract more image information. The convolutional layer with a stripe of 2 can prevent the loss of spatial information caused by the pooling layer. The power line itself is very slender, and the spatial information is particularly important for the identification and extraction of power lines.

**Table 1.** The convolutional CapsNet structure.

|  | Filter | Kernel Size | Stride | Output |
|---|---|---|---|---|
| input |  |  |  | 128 × 128 × 1 |
| Conv1 | 32 | 5 × 5 | 2 | 64 × 64 × 32 |
| Conv2 | 64 | 5 × 5 | 2 | 32 × 32 × 64 |
| Conv3 | 128 | 9 × 9 | 2 | 16 × 16 × 128 |
| primary capsule | 32 × 8 | 9 × 9 | 1 | 16 × 16 × 256 |
| digital capsule | - | - | - | 16 × 2 |
| output | - | - | - | 2 |

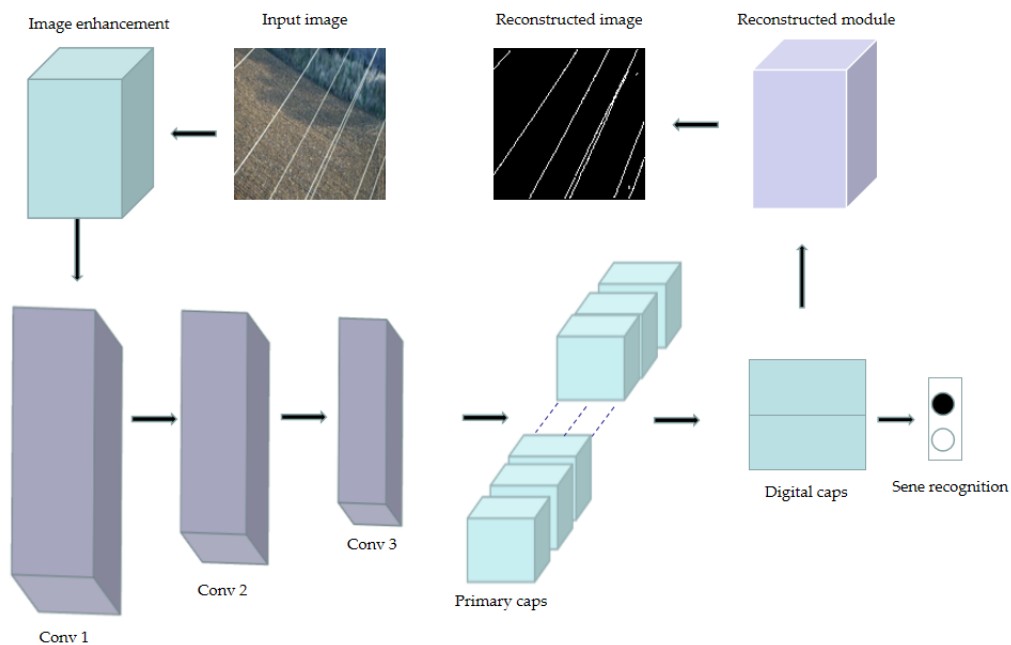

**Figure 4.** The proposed PLSR framework.

## 4. Scene Recognition Results and Analysis

### 4.1. Dataset and Experimental Configuration

The public data set of power line scenarios is adopted for experiment in this paper [9]. The dataset contains two subsets, infrared (IR) and visible light (VL). Each subset contains two parts, including and excluding. Each part has 2000 images of power line scenarios with $128 \times 128$ pixels. The subset with visible light [9] is used to carry out the experiment in this paper. The dataset is divided into training set, cross-validation set, and test set according to 3:1:1.

The configuration used in this paper, in terms of the hardware and the software platform, is shown in Table 2.

**Table 2.** Configuration of the experimental environment.

| Platform | Configuration |
|---|---|
| Operating system | 64 bit version of Windows 10 |
| Central processing unit (CPU) Graphic processing unit (GPU) | Intel(R) Core(TM) i9-10900k CPU @ 3.70 GHz NVIDIA GeForce RTX 2070 8 G |
| Deep learning framework | PyTorch1.7 |
| Compilers | PyCharm |
| Scripting language | Python 3.7 |
| Solid state disk (SSD) | 500 GB |

The experimental parameters used to train the proposed network are shown in Table 3.

**Table 3.** Experimental parameters of the convolutional CapsNet.

| Parameters | Configuration |
|---|---|
| Input Size | $128 \times 128 \times 1$ |
| Batch size | 64 |
| Optimizer | Adam |
| Learning rate | 0.001 |
| Training epochs | 200 |

*4.2. Evaluation Metric*

In this experiment, the accuracy rate is selected as the evaluation criteria, and the formula is given as Equation (9).

$$Accuracy = \frac{Number\ of\ correct\ predictions}{Total\ number\ of\ predictions} \tag{9}$$

The PLSR task is a binary classification problem, and the above-mentioned formula can be written as Equation (10).

$$Accuracy = \frac{(TP + TN)}{(TP + FP + TN + FN)} \tag{10}$$

where *TP* indicates that the actual case is positive, and the prediction is positive; *TN* indicates that the actual case is negative, and the prediction is negative; *FP* indicates that the actual case is negative, and the prediction is positive; *FN* indicates that the actual case is positive, and the prediction is negative.

*4.3. Experimental Results and Analysis*

4.3.1. Scene Recognition Results and Analysis

The visualization results of the proposed convolutional capsule network, with image enhancement on the visible light data set, are shown in Figure 5, where all the 32 images are visually displayed. The lower left part with the red font represents the real label, and the lower right part with the yellow font represents the model prediction results. Where 0 represents the scene without power lines, and 1 represents the scene containing power lines. All the 32 images are visually displayed, the presence or absence of power lines are correctly judged by using the proposed method.

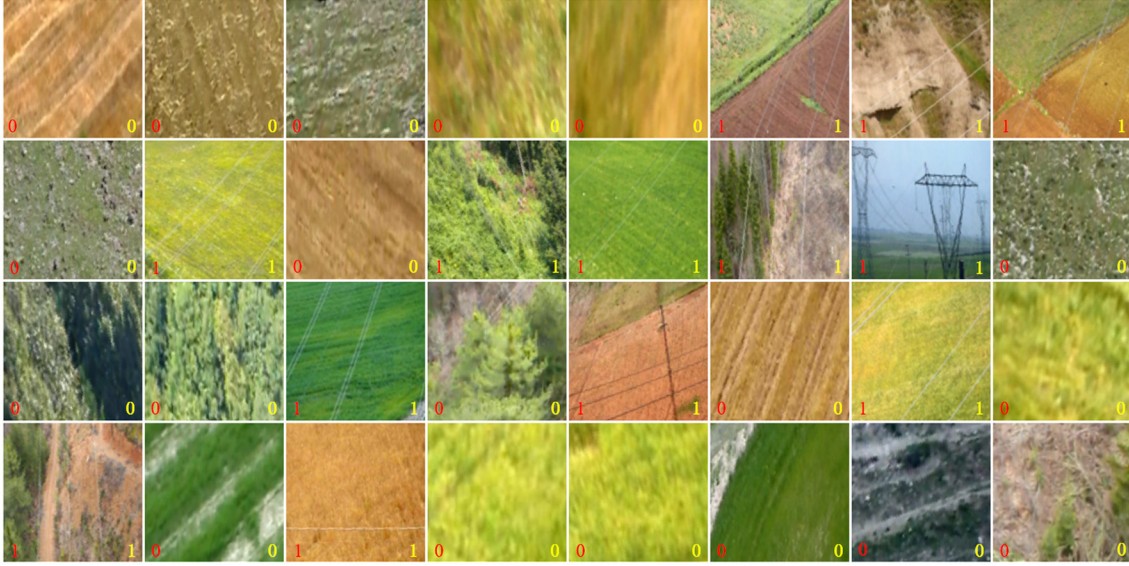

**Figure 5.** Visualization results of power line scene recognition.

In order to verify the superiority of the method on the visible light data set, the comparative experiments with the traditional image processing based methods [13,14] are given in Figure 6. The parameters of these compared methods are given based on literature [13,14]. The detailed methods are listed as follows: SVM is used to classify local binary pattern (LBP) features; naïve bayes (NB) is used to classify LBP features; random forest is used to classify LBP features; SVM is used to classify histogram of oriented gradient (HOG) features; naïve bayes is used to classify HOG features; random forest (RF) is used to

classify HOG features; SVM is used to classify classical selection DCT (CS_DCT) features; naïve bayes is used to classify CS_DCT features; random forest is used to classify CS_DCT features; SVM is used to classify reversed selection DCT (RS_DCT) features; naïve bayes is used to classify RS_DCT features; and random forest is used to classify RS_DCT features. Although a good detection result can be obtained by the DCT+RF, the feature extractor and matching method should be manually set. If the DCT+RF is tested on a larger dataset with a more complex background, the calculation will become more complicated, and the detection accuracy will not be guaranteed. The proposed model achieved the highest accuracy of 97.43%, which was 7.93% higher than the second place. It can be seen that on the visible light dataset, the proposed model has significant advantages over traditional image processing methods.

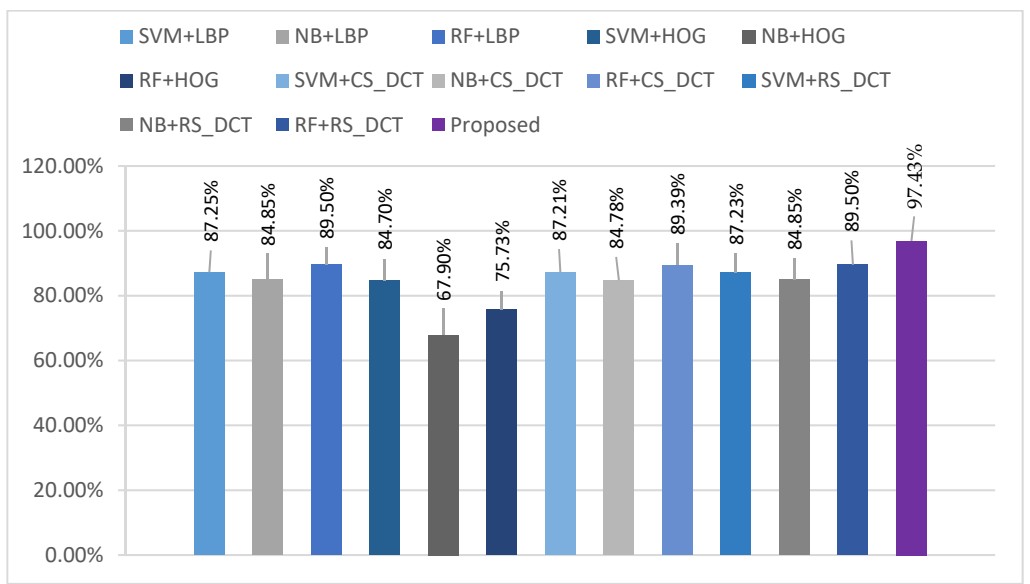

**Figure 6.** Comparison results with the traditional image processing based methods.

The proposed model is also compared with the deep learning methods implemented by us, and the experimental results are shown in Figure 7. The CapsNet is implemented as follows: After graying the $128 \times 128 \times 3$ power line scene image, it is resized to the size of $28 \times 28 \times 1$ and input into the original CapsNet network, without changing the network architecture. The accuracy is 77% by using the original CapsNet. The attention mechanism based CapsNet achieved accuracy of 78.8% [41]. Resizing the size from $128 \times 128 \times 1$ to $28 \times 28 \times 1$ simply results in the loss of the spatial information of power lines. Even with the attention mechanism-based CapsNet, it is hard to improve the accuracy of classification. Comparing the experimental results of the convolutional CapsNet, it can be seen that the two additional convolutional layers, without pooling operation, are effective, as the accuracy gets to 92.38% from 77%. The accuracy of the convolutional attention-based CapsNet (CA-CapsNet) reaches 93.5%. When image enhancement is added, the proposed model achieves the highest accuracy of 97.43%, and the guided convolutional attention-based CapsNet (GCA-CapsNet) obtains 97.15%. Since the power lines are very thin and run throughout the image, it is hard to design which part should be paid more attention, especially when both the edge lines of power lines and surrounding backgrounds are enhanced together.

Furthermore, U-net gets a very good classification performance, the accuracy of which is calculated by us from the result in [22]. It is verified that image enhancement with the guided filter is effective in improving the accuracy of the convolutional CapsNet and its variations. It also can be combined with other methods. It also has a further research value to improve the performance of itself by exploring more information.

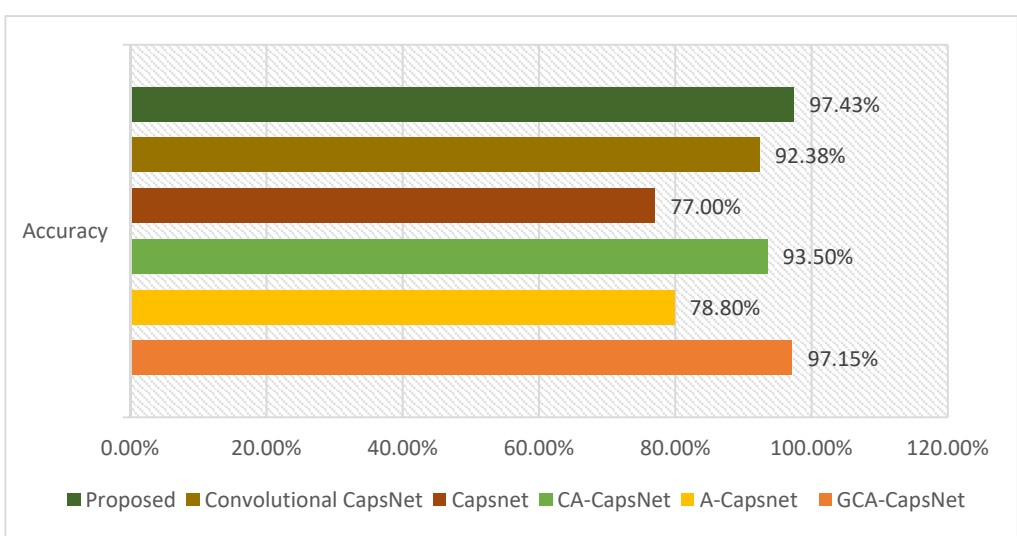

**Figure 7.** Comparison results with the deep learning based methods.

### 4.3.2. Performance Robustness Analysis

The robustness of the proposed PLSR method is tested in this section. The test dataset, containing 800 images, is selected for the experiment. The quantitative results are shown in Table 3. The accuracy of power line scene recognition in fog, snowfall, strong light, and motion-blurred scenes are 95.8%, 92.1%, 96.6%, and 88.3%, respectively. Compared with the normal scenes, the deviation of power line scene recognition accuracy in the above four scenes is −1.67%, −5.47%, −0.85%, and −9.37%, respectively. The deviation of motion-blurred scenes is slightly higher, but it is also less than 10%, and its performance is better than that of many normal scenes in Table 4. Other scenarios have a good performance robustness. Because the power line has the characteristics of small targets and weak features in aerial images, motion blur will affect the boundary response of the foreground and background. Through image feature enhancement and two additional convolution layers, the proposed method improves the robustness of power line scene recognition in the complex environments.

**Table 4.** Performance comparison of PLSR methods.

| Scenes | Accuracy |
|---|---|
| Foggy | 95.8 |
| Strong light | 92.1 |
| Snow fall | 96.6 |
| motion blur | 88.3 |

### 4.3.3. Generalization Test and Analysis

In order to evaluate the generalization performance of the proposed model more clearly, the test dataset in [12], containing 120 power line scene images with complex backgrounds, are selected, and another similar 80 images without power lines are also selected for testing. The total accuracy is 94.8%. Part of the test results of the proposed PLSR is shown in Figure 8. The lower left part with the red font represents the real label, and the lower right part with the yellow font represents the model prediction results. Where 0 represents the scene without power lines, and 1 represents the scene containing power lines. The experiment shows the recognition cases of eighteen images, of which the 13th image, the 14th image, and the 18th image are the display of false recognition cases.

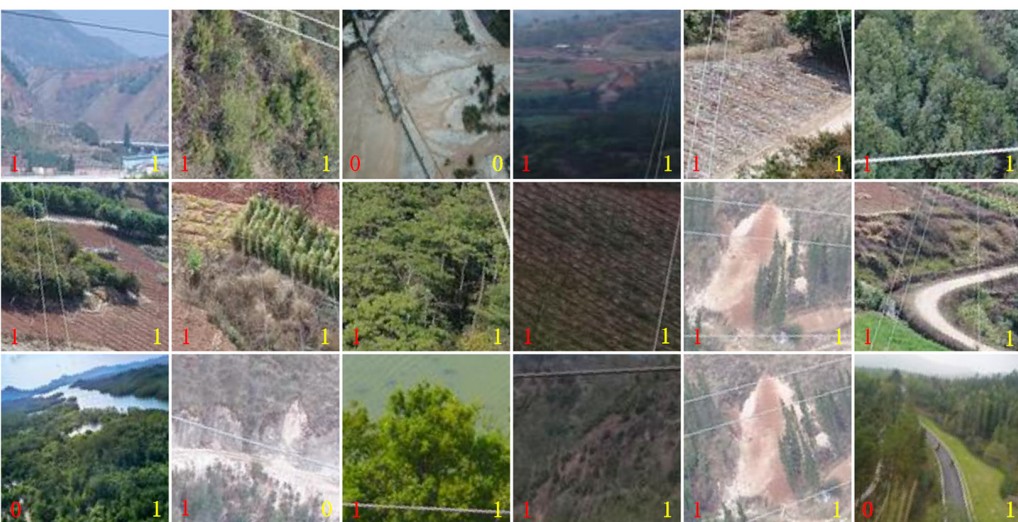

**Figure 8.** The visual generalization test results of the proposed method.

## 5. Reconstruction Results and Analysis

The CapsNet uses an automatic encoder structure to reconstruct data; the automatic encoder is composed of an encoder and decoder [16]. This section discusses and analyzes the effect of power line reconstruction based on capsule network. In the proposed CapsNet, the encoder is composed of a convolution layer, primary capsule layer, and digital capsule layer. The decoder includes three full connection layers. The decoder uses the image features of the power line scene generated in the encoder to reconstruct an image with the same size as the input image. During reconstruction, the encoder uses the difference of the mean square error between the reconstructed image and the label image. Low error indicates that the reconstructed image is similar to the label image. The decoder structure of the proposed model is shown in Figure 9.

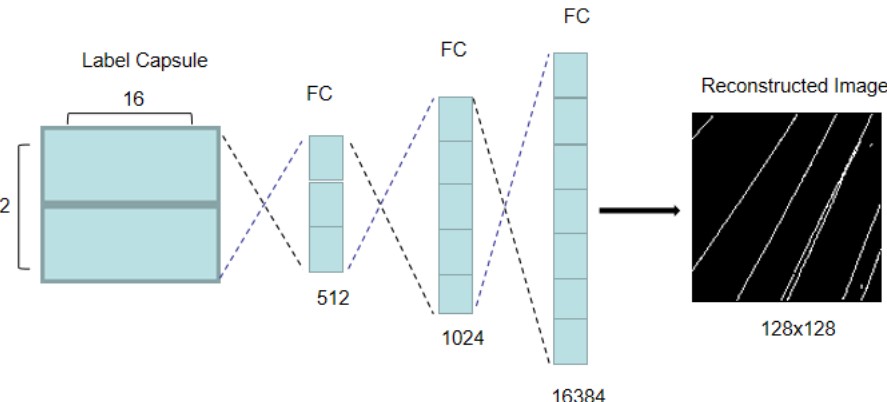

**Figure 9.** The decoder structure of the proposed model.

Based on the proposed method, the PLE results, with typical background, are shown in Figure 10. The six power line scene images with typical backgrounds are given in Figure 10a. The first and third pictures show the background of the tower. The second and fifth images show the field backgrounds. The fourth shows the grassland background. The sixth picture shows the road background. Figure 10b shows the real power line label corresponding to the original image, and Figure 10c shows the PLE results based on the proposed method. It can be seen that the power line can be completely extracted from the background by using the proposed model.

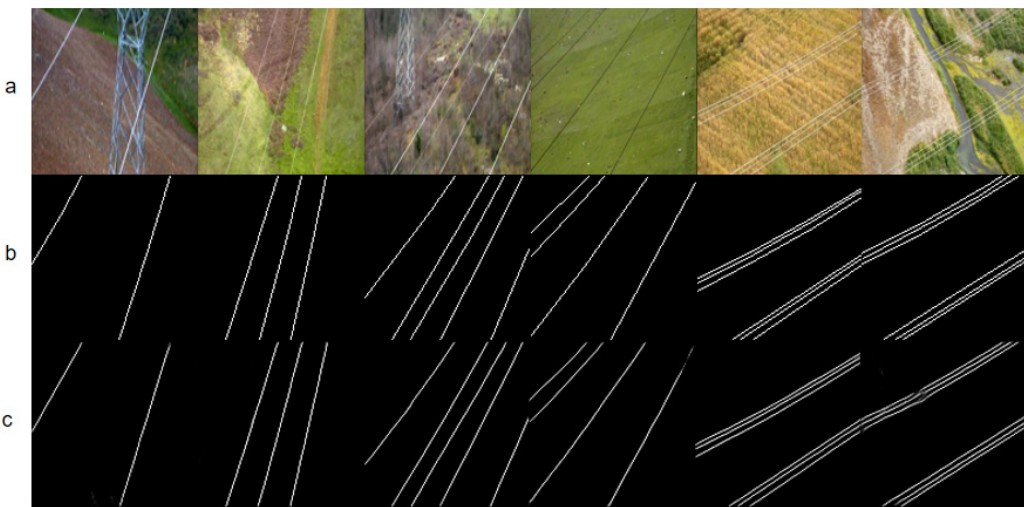

**Figure 10.** The PLE results with typical background. (**a**) Power line images. (**b**) Ground truth labels. (**c**) PLE results based on the proposed model.

In order to continue to evaluate the effect of the reconstruction model in the pixel level-recognition of power lines, Figure 11a shows six power line scene images with complex backgrounds. Due to the influence of complex backgrounds, the power lines are difficult to be found with naked eyes. The first and second pictures show the forest backgrounds. The third and fourth pictures show the mountain backgrounds. The fifth and sixth images show the backgrounds of the field. Figure 11b shows the real power line label corresponding to the original image, and Figure 11c shows the PLE results based on the proposed method. In these six images, although the power line is difficult to distinguish with naked eyes, the first image is perfectly extracted. The second and fourth images are partially bent and broken. The third and fifth pictures are partially missed, and the sixth picture has a small section of trees with multiple inspections. Overall, a good pixel level-recognition effect is achieved.

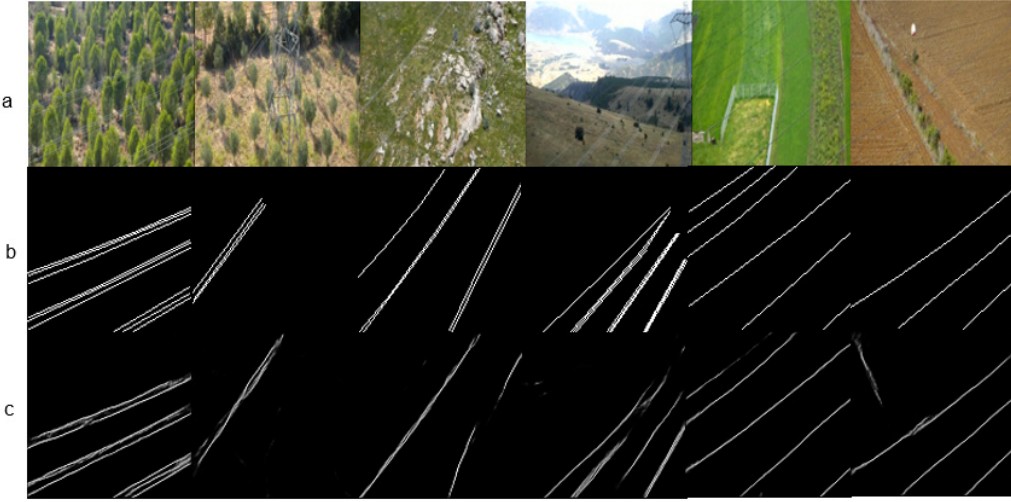

**Figure 11.** The power line extraction results under a complex background. (**a**) Power line images. (**b**) Ground truth labels. (**c**) PLE results based on the proposed method.

## 6. Conclusions

In this paper, the background of power line scene recognition is carefully analyzed at first, and the guided filter is found that can enhance the power line features effectively. Thus, the feature enhancement module with the guided filter is introduced to weaken

the influence of complex background images on power line detection and extraction. A convolutional capsule network is used to design the power line scene recognition and extraction method. Experiments show that the proposed method has a high recognition accuracy and good robustness in the PLSR task. The image output from the convolutional capsule network decoder can also obtain a better power line pixel-level recognition effect. Based on the proposed method, we can not only judge whether there is a power line scene, but also extract the power line completely from the scene image of power lines. It lays a foundation for the future research of UAV tracking along the line and fault diagnosis attached to components of power lines.

For the issue of not-so-perfect performance robustness in a strong light environment, the fusion of infrared images and visible light images can be introduced in the future, since in the strong light environment, although the power lines are indistinguishable from the background, the high-temperature power lines can be distinguished from the low-temperature background environment. For the issue of not-so-good performance robustness in a motion blur environment, in the future, more stable and active disturbance rejection UAV trajectory-tracking methods can be studied to obtain a better image capture effect and reduce motion blur in aerial images.

This article makes sense despite its simplicity. The selection of guided images in the guided filter is variable, which makes the combination with deep learning have unlimited potential. New features, such as edge detection, texture preservation, and image enhancement could be used as guiding images, which will enhance the performance of the network. In addition, the design is flexible and simple, and the computational complexity is lower than that of the attention mechanism, which can be widely combined without various deep learning tasks. In supervised learning, selecting the ground truth label as the guide image can greatly improve the training performance of the network. In unsupervised learning and predictive analysis tasks, first selecting the original image or enhanced image as the guided image, and then selecting the reconstructed image as the guided image as the relevant feedback will improve the performance of the image classification task.

**Author Contributions:** Conceptualization, methodology, writing—review and editing, supervision, project administration, K.Z.; validation, software, writing—original draft, visualization, S.Z.; formal analysis, investigation, resources, data curation, Z.J. All authors have read and agreed to the published version of the manuscript.

**Funding:** This research received no external funding.

**Data Availability Statement:** The data that support the findings of this study are openly available in [Mendeley Data] at [https://data.mendeley.com/datasets/n6wrv4ry6v/8 (accessed on 30 July 2022)] and [Mendeley Data] at [https://data.mendeley.com/datasets/twxp8xccsw/9 (accessed on 30 July 2022)]. The data partly support for generalization test are openly available in [Github] at [https://github.com/SnorkerHeng /PLD-UAV (accessed on 30 July 2022)].

**Conflicts of Interest:** The authors declare no conflict of interest.

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
