# Peer review of "Power Line Scene Recognition Based on Convolutional Capsule Network with Image Enhancement"

_electronics, doi:10.3390/electronics11182834_

Round 1
Reviewer 1 Report
As the title of the manuscript emphasizes, the work introduces an algorithm for Power Line Scene Recognition (PLSR) based on Capsule Network. It seems that the work is suitable for publication in a scientific journal. However, the quality of the text must be improved. There are many problems in the text that make it difficult to read and understand. An incomplete list of issues that need to be fixed is presented below.
(a) Acronyms need to be identified – for instance, what do LBP and HOG (line 48) mean?
(b) In the Introduction, the description of Yetgin et al. work is confused. First, the manuscript mentions that this work provided several PLSR methods. After that, the manuscript describes only one method.
(c) There are points where the manuscript makes claims that need references – for instance: (i) line 56 (“(…) some researches have tried to apply CNNs to PLSR.”), (ii) line 69 (“(…) which has been concerned and studied by more and more people.”)
(d) In line 92, the manuscript reports that a PLSR method based on CapsNet fused with image enhancement is improved in the work. If this method is being improved in the present work, it has already been proposed. Is it right?
(e) There are many typos, such as the dot in the final of the line 47 and in the words “information and”, in line 101. Even in equations, such as in (6), there are typos (missing parentheses).
(f) Following a different format than usual, the equations are written independently of the text.
(g) There are sentences that should be revised, such as the one which starts in line 85.
In short, it seems that the work is meritorious but the text must be completely reviewed.
Author Response
Reviewer 1.
As the title of the manuscript emphasizes, the work introduces an algorithm for Power Line Scene Recognition (PLSR) based on Capsule Network. It seems that the work is suitable for publication in a scientific journal. However, the quality of the text must be improved. There are many problems in the text that make it difficult to read and understand. An incomplete list of issues that need to be fixed is presented below.
Reply: Thank you very much for your earnest work and your valuable suggestions. We revise our manuscript substantially as your suggestions At the same time, inspired by your suggestions, we draw inferences from one instance and make major changes to the full text. See the red part of the resubmitted manuscript for details. I will reply to the specific opinions one by one as follows.
(a) Acronyms need to be identified – for instance, what do LBP and HOG (line 48) mean?
Reply: All the Acronyms are given the full name, for example, from line 284 to line 289.
(b) In the Introduction, the description of Yetgin et al. work is confused. First, the manuscript mentions that this work provided several PLSR methods. After that, the manuscript describes only one method.
Reply: This is revised, from line 49 to line 60. They proposed DCT features, but they also used LBP, HOG features, and SVM, Naive Bayes, and Random Forest for feature matching.
(c) There are points where the manuscript makes claims that need references – for instance: (i) line 56 (“(…) some researches have tried to apply CNNs to PLSR.”), (ii) line 69 (“(…) which has been concerned and studied by more and more people.”)
Reply: This is revised, meanwhile, other related parts are cited.
(d) In line 92, the manuscript reports that a PLSR method based on CapsNet fused with image enhancement is improved in the work. If this method is being improved in the present work, it has already been proposed. Is it right?
Reply: It it revised from line 98 to line 99. both the CapsNet and the improved one- Convolutional CapsNet are not proposed. For clear it, ‘improved CapsNet’ is changed to ‘ Convolutional CapsNet’ in the revised manuscript.
(e) There are many typos, such as the dot in the final of the line 47 and in the words “information and”, in line 101. Even in equations, such as in (6), there are typos (missing parentheses).
Reply: All the same questions are revised.
(f) Following a different format than usual, the equations are written independently of the text.
Reply: Equations are written by using the Math Type 6.
(g) There are sentences that should be revised, such as the one which starts in line 85.
Reply: all the sentences are revised.
In short, it seems that the work is meritorious but the text must be completely reviewed.
Reply: Other revised parts denoted as red colors.

Reviewer 2 Report
Paper deals with very important task. The authors improved deep learning PLSR method of complex background.
Paper has scientific novelty and great practical value.
It has a logical structure all necessary sections. The paper is technically sound. The experimental section is very good.
The proposed approach is logical, results are clear.
Suggestions:
1. The abstract section should be extended using numerical results obtained in the paper
2. The introduction section should be extended using more clearly the motivation of this paper.
3. It would be good to add the remainder of this paper at the end of the Introduction section
4. The related works part of the Introduction section should be extended using these papers: DOI:10.1007/978-3-319-63754-9_25, DOI: 10.1109/STC-CSIT.2015.7325423 among others
5. The authors should add all optimal parameters for all investigated methods
6. The conclusion section should be extended using: 1) numerical results obtained in the paper; 2) limitations of the proposed approach; 3) prospects for future research.
7. Some of references are outdated. Please fix it using 3-5 years old papers in high-impact journals.
Author Response
Reviewer 2:
Paper deals with very important task. The authors improved deep learning PLSR method of complex background.
Paper has scientific novelty and great practical value.
It has a logical structure all necessary sections. The paper is technically sound. The experimental section is very good.
The proposed approach is logical, results are clear.
Reply: Thank you very much for your earnest work and your valuable suggestions. We revise our manuscript substantially as your suggestions At the same time, inspired by your suggestions, we draw inferences from one instance and make major changes to the full text. See the red part of the resubmitted manuscript for details. I will reply to the specific opinions one by one as follows.
Suggestions:
- The abstract section should be extended using numerical results obtained in the paper
Reply: it is revised from line 19 to line 21.
- The introduction section should be extended using more clearly the motivation of this paper.
Reply: It is revised from line 42 to line 50.
- It would be good to add the remainderof this paper at the end of the Introduction section
Reply: It is revised from line 107 to line 111.
- The related works part of the Introduction section should be extended using these papers: DOI:10.1007/978-3-319-63754-9_25, DOI: 10.1109/STC-CSIT.2015.7325423 among othersLearning-based image scaling using neural-like structure of geometric transformation paradigm. Adv Soft Comput Mach Learn Image Process.
Reply: We downloaded all the papers, read them and get some special thoughts based on our method. Then revise the part from line 44 to line 97.
- The authors should add all optimal parameters for all investigated methods.
Reply: We make clear in the revised manuscript. for methods have codes, the parameters as the reference paper; for methods we do not implemented, but used the same dataset, the results are from their published papers. For the method we implement, we give the parameters.
- The conclusion section should be extended using: 1) numerical results obtained in the paper; 2) limitations of the proposed approach; 3) prospects for future research.
Reply: For numerical results, accuracy is added in the abstract, other results are not generalized used, It is better in words. Limitations are added from line 406 to line 413. future research is added from line 414 to line 424.
- Some of references are outdated. Please fix it using 3-5 years old papers in high-impact journals.
Reply: This part is revised from line 438 to line 448, line 498 to line 512.

Reviewer 3 Report
Overall, I think this paper is not very innovative. First, image enhancement method (guided filtering) is an existing method. Secondly, CapsNet is also a quite common network. Although the author shows that the structure of CapsNet has been improved, there is no convincing improvement point from Figure 2. Furthermore, the experimental comparison is relatively simple, and the compared methods and means are very basic. Since the network structure is too simple, there are no ablation experiments. The visualization results are also not compared with other methods.
In short, this paper is relatively simple as a whole, and needs to be greatly improved in terms of theoretical analysis, network design, network structure, experimental data and comparison of results.
Author Response
Reviewer 3:
Overall, I think this paper is not very innovative. First, image enhancement method (guided filtering) is an existing method. Secondly, CapsNet is also a quite common network. Although the author shows that the structure of CapsNet has been improved, there is no convincing improvement point from Figure 2. Furthermore, the experimental comparison is relatively simple, and the compared methods and means are very basic. Since the network structure is too simple, there are no ablation experiments. The visualization results are also not compared with other methods.
In short, this paper is relatively simple as a whole, and needs to be greatly improved in terms of theoretical analysis, network design, network structure, experimental data and comparison of results.
Reply: Thank you very much for your earnest work and your valuable suggestions. We revise our manuscript substantially as your suggestions At the same time, inspired by your suggestions, we draw inferences from one instance and make major changes to the full text. See the red part of the resubmitted manuscript for details. I will reply to the specific opinions one by one as follows.
- The public dataset for power line extraction and scene recognition is few, only one can be used for the scene recognition part. We explain from line 43 to line 48.
- Few published papers focus the research on scene recognition of power lines, no complex methods proposed. Thus, we are trying to improve the commonly used method at first, then study methods with theoretical analysis. We explain from line 49 to line 76.
- This article makes sense despite its simplicity. The selection of guided images in the guided filter is variable, which makes the combination with deep learning have unlimited potential. New features such as edge detection, texture preservation, and image enhance-ment could be used as guiding images, which will enhance the performance of the net-work. In addition, the design is flexible and simple, and the computational complexity is lower than that of the attention mechanism, which can be widely combined without var-ious deep learning tasks. In supervised learning, selecting the ground-truth label as the guide image can greatly improve the training performance of the network. In unsuper-vised learning and predictive analysis tasks, first selecting the original image or enhanced image as the guided image, and then selecting the reconstructed image as the guided im-age as the relevant feedback will improve the performance of the image classification task.We explain from line 179 to line 220, line 406 to line 425.
- Other revised parts denoted as red colors.

Round 2
Reviewer 1 Report
The manuscript was substantially revised, and the quality of the text was improved. There are still parts of the text that needed to be corrected (examples: (a) the sentence “Second, for feature extraction.” in line 58, and (b) equation (6) with missing symbols). However, the contribution of the work is clear. I agree with the authors when they state in the conclusion that the “article makes sense despite its simplicity”. In my opinion, despite the simplicity of the proposal, the topic is relevant, and the results are interesting. Therefore, my overall recommendation is to accept after minor revision.
Author Response
Thank you very much for your earnest work and your valuable suggestions. We revise our manuscript substantially as your suggestions again. See the annotated part of the resubmitted manuscript for details. Both (a) and (b), and the related parts are corrected. We also improved the English expression.
Reviewer 3 Report
I still think that the innovation of the article is limited. The author also mentioned the attention mechanism in the response letter, so I suggest that the author consider the attention mechanism when improving the network. Although the modified version adds more experiments, the compared deep learning networks are all commonly used backbone networks.
Author Response
Thank you very much for your earnest work and your valuable suggestions. We revise our manuscript substantially as your suggestions again. See the annotated part of the resubmitted manuscript for details. I will reply to the specific opinions one by one as follows.
- The theoretical innovation of this manuscript is really limited, but we are not just combine A with B, we analysis the inherent features of power lines (from line 149 to line 157), and find the potential useful methods step by step. The motivation of our research is added in part 3.1. (line 148). Furthermore, the existed methods compared in this paper, are published within 5 years, or is not published, just implemented by us.
- Deep learning in power line scene recognition just began, there are fewer publicly available datasets, but we believe more datasets will be made public in the future. This research is just a door to combine guided filter and deep learning, we are doing a series of innovative studies based on this field.
- Attention mechanism in [41] published in 2022, is used for comparison, from line 322 to line 331. Other people maybe just make an improvement and proposed an attention mechanism based method for possibility publish in this journal. We do not choose that, because we do not know the detail mechanism how the power lines are paid more attention. In fact, it is really a hard work. We will spend more time making the attention mechanism focus to the power lines, instead of reinforcing all the edge features. We got better results than attention mechanism method, and our method has a more widely improvement prospect. In deference to the research, I think our work is also meaningful.